# Differential Signaling Profiles of MC4R Mutations with Three Different Ligands

**DOI:** 10.3390/ijms21041224

**Published:** 2020-02-12

**Authors:** Sarah Paisdzior, Ioanna Maria Dimitriou, Paul Curtis Schöpe, Paolo Annibale, Patrick Scheerer, Heiko Krude, Martin J. Lohse, Heike Biebermann, Peter Kühnen

**Affiliations:** 1Institute of Experimental Pediatric Endocrinology, Charité–Universitätsmedizin Berlin, corporate member of Freie Universität Berlin, Humboldt-Universität zu Berlin, and Berlin Institute of Health, D-10117 Berlin, Germany; sarah.paisdzior@gmx.de (S.P.); ioanna-maria.dimitriou@charite.de (I.M.D.); heiko.krude@charite.de (H.K.); heike.biebermann@charite.de (H.B.); 2Max Delbrück Center, Robert-Rössle-Straße 10, 13092 Berlin, Germany; Institute of Pharmacology and Toxicology, University of Würzburg, D-97078 Würzburg, Germany; paul.schoepe@charite.de (P.C.S.); paolo.annibale@mdc-berlin.de (P.A.); Martin.Lohse@mdc-berlin.de (M.J.L.); 3Group Protein X-ray Crystallography and Signal Transduction, Institute of Medical Physics and Biophysics, Charité- Universitätsmedizin Berlin, corporate member of Freie Universität Berlin, Humboldt-Universität zu Berlin, and Berlin Institute of Health, D-10117 Berlin, Germany; patrick.scheerer@charite.de

**Keywords:** Melanocortin 4 receptor (MC4R), Melanocyte stimulating hormones MSH, G protein coupled receptor (GPCR), biased signaling

## Abstract

The melanocortin 4 receptor (MC4R) is a key player in hypothalamic weight regulation and energy expenditure as part of the leptin–melanocortin pathway. Mutations in this G protein coupled receptor (GPCR) are the most common cause for monogenetic obesity, which appears to be mediated by changes in the anorectic action of MC4R via G_S_-dependent cyclic adenosine-monophosphate (cAMP) signaling as well as other signaling pathways. To study potential bias in the effects of MC4R mutations between the different signaling pathways, we investigated three major *MC4R* mutations: a G_S_ loss-of-function (S127L) and a G_S_ gain-of-function mutant (H158R), as well as the most common European single nucleotide polymorphism (V103I). We tested signaling of all four major G protein families plus extracellular regulated kinase (ERK) phosphorylation and β-arrestin2 recruitment, using the two endogenous agonists, α- and β-melanocyte stimulating hormone (MSH), along with a synthetic peptide agonist (NDP-α-MSH). The S127L mutation led to a full loss-of-function in all investigated pathways, whereas V103I and H158R were clearly biased towards the G_q/11_ pathway when challenged with the endogenous ligands. These results show that *MC4R* mutations can cause vastly different changes in the various MC4R signaling pathways and highlight the importance of a comprehensive characterization of receptor mutations.

## 1. Introduction

A complex hypothalamic network of orexigenic and anorexigenic signals regulates body weight and energy expenditure. One of the key players in this network is a G protein coupled receptor (GPCR), the melanocortin 4 receptor (MC4R), which is mainly expressed in the paraventricular nucleus (PVN) of the hypothalamus. The MC4R receives input from the leptin–melanocortin pathway and thereby integrates peripheral and central metabolic signals. Activation of the MC4R by the proopiomelanocortin (POMC)-derived melanocyte stimulation hormones alpha and beta (α-MSH, β-MSH) counteracts the signal of orexigenic peptides such as neuropeptide Y (NPY), Agouti-related peptide (AgRP) or ghrelin, and therefore deceases appetite [1,2,3]. Its pivotal role in weight homeostasis became obvious when mutations in the *MC4R* gene were identified in obese patients [4,5].

The MC4R signals mainly via activation of the stimulating G protein G_S_, which leads to an increase in intracellular cyclic adenosine-monophosphate (cAMP) [6]. In recent years, more and more studies provided evidence that the MC4R is able to couple to a variety of signaling pathways, such as extracellular regulated kinase (ERK) activation [7,8,9], and to other G proteins, such as G_i/o,_ G_q/11_ and G_12/13_ [10,11,12,13]. These pathways might also be involved in the anorexigenic effect of the activated MC4R [14]. The receptor exhibits ligand-independent constitutive activity (basal activity) [15] and is able to form homodimers as well as homo-oligomers [16]. The endogenous peptide AgRP has been described to reduce this constitutive activity, i.e., to act as an inverse agonist [15,17] or as a competitive antagonist [15,18]. In addition to MC4R activation by endogenous ligands, the MC4R can be activated by a variety of synthetic peptide ligands, such as (Nle^4^, d-Phe^7^)-α-MSH (NDP-α-MSH) [19].

So far, many mutations identified in obese patients have been functionally characterized in overexpressing in vitro systems, and a number of these mutations showed a loss-of-function in G_S_ signaling. However, other *MC4R* mutations were described as wild type (WT)-like and were, therefore, believed to be irrelevant for the obese patient phenotype.

In recent years, several *MC4R* mutations have been identified that exhibit biased signaling, which is defined as the selective activation or inactivation of one pathway over another [20]. Tao and colleagues were the first to describe biased signaling of *MC4R* mutations in the mitogen-activated protein kinase (MAPK) pathway. Interestingly, in some of the tested mutations no functional defect in G_S_ signaling had been described before [21]. Recently, bias towards β-arrestin has been studied, revealing mutations with unremarkable changes in G_S_ signaling but with a markedly affected ability to recruit β-arrestins [22,23].

These observations indicate that it may be necessary to study more than just the receptor’s G_S_-signaling properties in order to fully characterize *MC4R* mutations. The aim of the current study was, therefore, the comprehensive characterization of three prototypical *MC4R* mutations in their ability to activate all four major G protein families, plus ERK activation, β-arrestin2 recruitment and ligand-dependent internalization. As a proof of principle study, we focused on a very common *MC4R* variant, V103I, and two *MC4R* missense mutations: S127L and H158R.

The most common single nucleotide polymorphism (SNP) in Europe, V103I, is located in the transmembrane domain (TMD) 2, and has been mostly described as WT-like in G_S_ signaling [24,25,26,27,28,29] and has been suggested to be protective against obesity [23,30,31,32,33,34,35,36,37]. The second *MC4R* mutation, S127L, has been frequently reported as a partial loss-of-function in G_S_ signaling, which has an increased constitutive activity and is located in TMD 3 [26,38,39,40,41,42,43,44,45]. The third investigated naturally occurring *MC4R* mutation, H158R, is a constitutively active gain-of-function mutation in G_S_ signaling that has a decreased ability to form homodimers [46,47]. It is located in the second intracellular loop (ICL2). Since V103I is the only *MC4R* variant, which is defined by a minor allele frequency (MAF) above one percent in the normal population, whereas the other two are considered to be rare with a lower MAF, we will continue referring to all of them with the term “mutations”.

With regard to the G_S_ activation pattern, these three mutations are vastly different. However, these findings do not strictly align with the phenotype. This suggests that other GPCR signaling pathways might be important for weight regulation, and therefore these were investigated in the present study.

## 2. Results

In this study, we characterize the three described *MC4R* mutations in vitro for all known/so far reported signaling pathways to explore the role of G_S_ and non-G_S_ pathways in weight regulation. Therefore, in addition to cAMP measurements to assess G_S_ activation in the absence and presence of 100 nM AgRP, we measured G_i/o_ activation via determination of second messenger (cAMP) modulation by adenylyl cyclase (AC) in the absence or presence of pertussis toxin (PTX) pretreatment. In the presence of PTX, which locks the Gα_i/o_ subunit in its GDP-bound inactivated state, the G protein is no longer able to inhibit the AC to form cAMP [48]. Although this is an indirect way of measuring the contributions from G_i/o_ signaling to the total amount of intracellular cAMP, we will refer to these measurements as G_i/o_ coupling in order to simplify the results part. We used reporter gene assays measuring phospholipase C (PLC) activity through nuclear factor of activated T-cell (NFAT) response for G_q/11_, RhoA activation via serum-response factor (SRF) for G_12/13,_ and ERK1/2 phosphorylation via the serum response element (SRE). Recruitment of β-arrestin2 by the receptor after the ligand challenge was determined by protein interaction studies and total and cell surface expression were analyzed using a protein complementation assay. In addition, ligand-induced internalization was visualized by staining the receptor with a fluorophore-coupled ligand.

### 2.1. Total and Cell Surface Expression of the Three MC4R Mutations

Cell surface and total expression of all three variants were determined. This provides information on protein folding and how the mutation influences the trafficking of the receptor to the plasma membrane, which may influence signaling. Introduction of these three mutations into the MC4R did not result in any significant changes in cell surface expression and total expression (Figure 1).

### 2.2. Internalization of the Three MC4R Mutations

In order to investigate the internalization ability of all three *MC4R* mutations, we used confocal imaging of labeled MC4R as well as β-arrestin2 recruitment as an indication of receptor desensitization. In our experience, MC4R C-terminally tagged with GFP is exhibiting signaling deficits, hence we chose another labeling approach. We used TAMRA-labeled NDP-α-MSH in order to stain receptors expressed at the cell surface and quantified the amount of labeled receptors in endosomes after 30 min of stimulation using confocal images (Figure 2). When compared with MC4R WT, we observed less endosomes containing V103I, whereas S127L and H158R showed an increased internalization (Appendix A). It should be noted that when the amount of DNA transfected and the image contrast were held stable for all confocal images, V103I showed stronger staining compared to the WT. This suggests that the internalization deficit of this mutation increases the amount of receptors on the cell surface after stimulation, although the cell surface expression of the mutation was not significantly increased in the absence of a ligand, as shown in Figure 1.

Interestingly, the basal β-arrestin2 recruitment of all three mutations was significantly reduced in comparison to the WT (Appendix A) with the strongest effect observed for V103I. When stimulated with the endogenous ligands, all three mutations had a significantly diminished recruitment, whereas NDP-α-MSH significantly increased the interaction with β-arrestin2 for all mutations in comparison to the MC4R WT.

### 2.3. Comprehensive Functional Characterization of the Three MC4R Mutations

A complete characterization of all known pathways for three receptor mutations for three different agonists generates a substantial amount of data (Figure 1, Appendix A, and Appendix A). The intention of this study was the evaluation of three mutations in comparison to WT signaling (determination of biased signaling) after a challenge with the two endogenous ligands and a highly potent (in G_S_ signaling) peptidic ligand. Therefore, an easy way to communicate biased signaling was used to present the results in a more comprehensive way, by plotting the bias calculated as described in the method section for each *MC4R* mutation. The values derived from concentration–response experiments (all concentration–response curves for all mutations and the three ligands are presented in Figure 3), which were fitted to non-linear regression models to calculate the efficacy of each ligand at the receptor mutation (E_max_) and the potency of the ligand at the receptor mutation (EC_50_). Values were normalized to the reference, *MC4R* WT, which is plotted as 1.0 for each pathway. Values above 1.0 represent an increase in signaling capacity in this pathway in comparison to the WT, whereas values below 1.0 represent a decrease in activity. Mutations with a WT-like behavior have bias values close to 1.0, meaning that there is in fact no bias. Efficacy, potency and bias values are stated in Appendix A.

#### 2.3.1. Stimulation with α-MSH Leads to a Strong Signaling Bias towards the G_q/11_ Pathway for MC4R Mutations V103I and H158R

Functional characterizations of *MC4R* mutations are usually performed with the endogenous agonist α-MSH in most published studies. We characterized MC4R mutations V103I, S127L and H158R in all mentioned pathways with this ligand and identified that all three mutations showed clear signaling biases towards particular pathways (Figure 4).

The mutation V103I exhibited a strong bias towards PLC activation via G_q/11_, where it clearly showed a gain-of-function. On the other hand, it had a decreased capacity to couple to G_i/o_. The data suggest a tendency towards a reduced antagonist effect of AgRP, which was, however, not significant. In every other tested pathway, the mutation signal was comparable to the WT after stimulation with α-MSH. This is in line with most published data regarding this mutation [24,25,26].

The loss-of-function mutant S127L was clearly impaired in every tested pathway when challenged with α-MSH. Treatment with AgRP showed a trend towards reduced antagonistic potency, which was again not significant.

We found the already-reported gain-of-function in G_S_ signaling for the H158R mutation with a decreased EC_50_ and an increase in maximal response (E_max_) [46]. Interestingly, the increase in bias values for G_S_ signaling in the presence of AgRP implies a loss-of-antagonist effect for the mutation. Notably, H158R showed a similar strong bias towards G_q/11_ as the V103I mutant with a rise in PLC activation. In other tested pathways, the mutation closely resembled the WT activity.

#### 2.3.2. Challenge with β-MSH Results in a Different Activation Pattern Compared to α-MSH for MC4R V103I and H158R Mutations

The second endogenous agonist for MC4R, β-MSH has been neglected in most MC4R studies so far, but is proven to be important for weight regulation [22,48,49] and was therefore included in this study (Figure 5).

Here, the V103I mutation presents itself with a distinctly different signaling behavior compared to α-MSH stimulation. Although it still exhibits an increased G_q/11_ coupling, its capacity to activate the G_S_ pathway is reduced (though not significantly), which is also reflected in its response to the antagonist AgRP. In addition, it showed a strong positive bias towards G_i/o_ coupling, RhoA activation via G_12/13_ and increased ERK phosphorylation. S127L on the other hand again presented full loss-of-function in all tested pathways and even its response to AgRP was impaired. The responses of H158R to β-MSH were comparable to its α-MSH responses with increased G_S_ and G_q/11_ signaling and WT-like activity in G_12/13_ and β-arrestin2 recruitment, though it also showed augmented ERK phosphorylation. Again, AgRP loses its antagonistic effect for this mutation. In contrast to α-MSH, H158R showed a strong bias towards G_i/o_ signaling when stimulated with β-MSH.

#### 2.3.3. MC4R V103I and H158R Lost Their bias Towards G_q/11_ After NDP-α-MSH Stimulation

NDP-α-MSH, a synthetic peptide analogue to α-MSH, is a very potent agonist [50] and its radiolabeled versions are suitable for in vitro testing of *MC4R* mutations. Since it is widely used, we included this ligand in our study (Figure 6).

In comparison to the endogenous ligands, the V103I mutant signaling characteristics after the NDP-α-MSH challenge showed yet again a different picture. Most notably, the bias towards G_q/11_ was absent and PLC activation was similar to WT. In addition, G_S_ signaling was increased in comparison to WT activity, which was then also reflected in its response to the antagonist AgRP. RhoA activation via G_12/13_ as well as ERK phosphorylation were increased, but the general picture of this mutation shows a signaling behavior that is very close to MC4R WT when stimulated with NDP-α-MSH. This potent agonist was not able to rescue the loss-of-function mutation S127L and its diminished signaling capacity remained, except for β-arrestin2 recruitment that exhibited a WT-like behavior. Lastly, MC4R H158R showed the typical gain-of-function in G_S_ signaling when challenged with NDP-α-MSH. Strikingly, this mutation also lost the strong bias towards PLC activation via G_q/11_, but exhibits clear bias towards G_12/13_ and ERK phosphorylation with increased signaling.

## 3. Discussion

To understand the underlying molecular mechanisms caused by mutations in GPCRs in general, a comprehensive functional characterization is essential. Based on the complexity of changes induced by genetic variants a proper characterization should include evidence on the following aspects: (1) How does a mutation influence the total receptor expression in the cell? (2) What is the impact of the mutations on cell surface expression? (3) Is there any signaling bias involving all known signaling pathways and endogenous ligands for this specific GPCR? (4) Does a mutation create any bias of ligand-induced G protein activation versus β-arrestin recruitment? (5) Does an altered response to the antagonist (in case of a known endogenous antagonist) exist? This extensive study can help to elucidate the role of each of these mechanisms in the physiological and potential pathophysiological role of a receptor and its mutations.

Furthermore, it will be of importance to establish benchmarks and guidelines concerning the way in which in vitro data of the functional characterization should be presented. Here, it would be relevant to present either raw data or calculated values normalized to WT basal activity (at equivalent expression levels) to ensure comparability of assay data. We recommend with respect to MC4R-related in vitro assays that EC_50_ should only be calculated if the maximal response is greater than 25% of the maximal WT activity. In addition, the use of different read-out systems for signaling pathways makes a comparison between different studies difficult, therefore a clear statement is necessary on how those data were determined.

Additionally, the choice of cell system may affect the results and should be considered carefully. Endogenous cell systems are always considered the best choice; however, for many studies those are not available. Chinese hamster ovarian (CHO) and African green monkey (COS) cell lines are easy to transfect and are able to produce high amounts of second messengers such as cAMP, which gives them an important role in GPCR research. For neuroendocrine receptors, hypothalamic rodent cell lines, such as GT1-7 or mHypoE-N39 and mHypoE-N41, are suitable due to their close physiological background, but with current options, transfection has been shown to be difficult. For all the mentioned cell lines, their non-human background complicates any actual interpretations on the physiological roles of human receptor function. Human cell lines such as human embryonic kidney cells (HEK293) are easy to transfect and suitable for a variety of functional assays and are well established in GPCR research. HEK293 cells are proven to not express MCRs [51] and do not respond to NDP-α-MSH [52]. We used this particular cell line in the presented study and are aware of the limitations of this cell system as it does not resemble the repertoire of proteins of hypothalamic neurons. For investigations of MC4R, it has to be noted that the melanocortin 2 receptor accessory protein 2 (MRAP2), a protein that has been shown to decrease constitutive activity of MC4R and also increase the responsiveness to α-MSH in G_S_ signaling [53], is marginally expressed in our used cell system [54]. This needs to be taken into consideration during interpretation of the data, especially regarding basal constitutive activity.

For a long time, ERK phosphorylation was considered as a G protein-independent and β-arrestin-dependent signaling pathway. Very recently, Grundmann et al. reported data that hint towards the need for G protein presence for ERK phosphorylation, whereas β-arrestins seem to be less involved [55]. Therefore, we would currently consider β-arrestin2 recruitment solely as part of the internalization machinery in the following discussion.

A recent genome-wide association study (GWAS) has reported that patients with pathogenic *MC4R* mutations had a very variable outcome depending on their genetic background. Here, polygenetic scores (PGS) were calculated for each individual, which measure the cumulative effect of common alleles and their effects on the person’s risk of disease. So, *MC4R* mutations carriers with favorable PGS had an average BMI, which was approximately equal to the population mean, while those with an unfavorable PGS were more strongly affected [56]. Therefore, we decided not to perform genotype–phenotype outcome correlations, since we assume that correlation of a patient’s phenotype with functional data would not result in robust data. In the following discussion we will focus on the interpretation of our findings from a functional standpoint and their comparison to published information about the tested *MC4R* mutations.

### 3.1. MC4R Mutations are Expressed Equally, but Internalize Differently

Our investigations regarding total and cell surface expression of the *MC4R* mutations showed no significant change in our chosen cell system in the basal state. Therefore, we can conclude that the results from the functional characterization reflect the actual signaling capacity of the mutations rather than their expression on the cell surface. This is in line with published data for V103I [57] and some reports for S127L when transfected transiently [39,40,58]. However there are conflicting results obtained in others studies, which showed a decreased cell surface expression of S127L, but no change in total expression in stably transfected cells [41].

In our study, the capacity to form endosomes after stimulation with NDP-α-MSH of V103I was disturbed, which aligns with recently published data [23]. This internalization deficit might be due to the significantly decreased β-arrestin2 recruitment we observed; however, this finding could not be recapitulated in a recent publication [23]. This discrepancy might be explained by the way the data is generated as we utilized a bioluminescence resonance energy transfer method (NanoBRET) in comparison to the split-luciferase approach of the previous study (Figure 3 and Appendix A). To investigate the conflicting evidence for V103I, future approaches could involve a kinetic measurement of β-arrestin2 recruitment as well as phosphorylation by GPCR kinases (GRKs).

S127L has been shown to become internalized similarly to the WT when stimulated with α-MSH, determined by ELISA studies [58]. We even found an increased amount of endosome formation after stimulation with TAMRA-NDP-α-MSH of this mutation, which is in line with increased β-arrestin2 recruitment after stimulation with the synthetic ligand.

### 3.2. Constitutive Activity in all Tested Pathways Differed for Each Mutation Compared to WT-MC4R

The MC4R has been shown to be constitutively active in the G_S_ pathway [15], and two out of the three investigated mutations have been reported to exhibit increased basal activity. For S127L, some publications identified a constitutive activity in G_S_ [58,59], which could not be determined in the present study. One reason might be that in our HEK293 cell system MRAP2 is expressed in low amounts, which might reduce basal signaling [22,53,60]. We did, however, reproduce the increase in basal G_S_ coupling that was reported for H158R [46].

Since only a few studies addressed other pathways for MC4R, information about basal activity is scarce. S127L has been investigated for ERK phosphorylation and was found to have an increased basal activity in this pathway [57]. Although we saw a tendency towards increased constitutive ERK activity for all tested *MC4R* mutations, the results were not significant. H158R, on the other hand, was found to have reduced basal ERK phosphorylation compared to MC4R WT [21], which we could not reproduce.

### 3.3. Signaling Bias towards G_q/11_ of Two Mutations When Stimulated with α-MSH

The endogenous ligand α-MSH is commonly used to investigate *MC4R* mutations in vitro. Its effect has been described for all three studied mutations in various details. G_S_ signaling determined in this study is in accordance with the majority of already published data for all three mutations. Stimulation by α-MSH in presence of 100 nM AgRP resulted in impaired antagonistic potency for S127L, but an increase for H158R. For *MC4R* S127L, AgRP had a reduced antagonistic potency, which is in line with published data [40]. One study suggests that V103I has a modestly reduced AgRP antagonistic potency, though this has only been tested with a synthetic agonist (MTII) [33]. We did notice a tendency towards a decreased antagonistic effect, which was mainly due to a low efficacy compared to the WT. For the H158R mutation, no additional data about the potency of the antagonist exists in the literature.

The most eminent observed effect was the increase in G_q/11_ signaling for two of our mutations, V103I and H158R, which have never been investigated in this pathway before. Over the years, evidence has accumulated that suggests a role of PLC activation in the anorexigenic effect of *MC4R*, and the activation of G_q/11_ signaling through MC4R has been documented in several studies [14,61,62,63]. We recently reported that naturally occurring *MC4R* mutations are found in obese patients with normal function in G_S_, but impaired NFAT activation as read-out system of G_q/11_ activation could be rescued in vitro with a synthetic ligand biased towards PLC activation [12]. This hints to the importance of the G_q/11_ pathway, as a strong bias towards PLC activation of V103I might play a part in its suggested protective role against obesity [30,31,32,33,34]. The rare mutation H158R is located in the ICL2, which is part of the G protein binding site and has been postulated to be important for G_q/11_ coupling as well [64]. The observed gain-of-function in this pathway therefore suggests functional relevance of this position.

Similarly, ERK phosphorylation has been suggested to be important for MC4R function in vitro and in vivo [7,8,9], though the physiological role of this pathway is poorly understood. MC4R triggered ERK activation can be modulated by AgRP [57] and the MAPK pathway has been shown to result from the activation of several G proteins, such as the Gi_/o_ and G_q/11_ proteins [65]. For V103I, ERK phosphorylation has been reported to be increased after stimulation with all three tested ligands [23]. Though we see a tendency towards an ERK bias, our study did not reveal a significant increase in activity for this pathway.

### 3.4. MC4R Mutations Respond Differently to the Endogenous Melanocortins

The endogenous ligands α- and β-MSH share a core amino acid sequence (His-Phe-Arg-Trp) but differ in length and protein stability. They are both involved in the anorexigenic effect of MC4R signaling [1,22,48,49], though the reason for two different peptides that convey a rather similar function within the pathway is not entirely clear. For the MC4R WT there are no strong differences in the signaling profile of the two ligands; however, G_S_ signaling of β-MSH in the presence of 100 nM AgRP is not as strongly affected compared to the one of α-MSH. Interestingly, the stimulation of V103I with β-MSH had a tendency towards low G_S_ signaling, which is conflicting with recently published results [23]. The S127L mutant again exhibited a loss-of-function in G_S_, as already published [40]. There is no published information about the stimulation of H158R with β-MSH, though the gain-of-function similar to α-MSH was expected.

For the other tested pathways, the two *MC4R* mutations V103I and H158R have a diverse signaling profile when stimulated with β-MSH compared to α-MSH. They still exhibit a strong bias towards PLC, though other signaling pathways are enhanced as well, such as G_i/o_ and G_12/13_ coupling. In addition, H158R was able to increase ERK phosphorylation in comparison to the WT. This might be the result of the strong bias towards G_q/11_ and G_i/o_ coupling that increased the activation of MAPK. As mentioned before, V103I was published to exhibit an increased ERK phosphorylation [23]. Again, in our cell system, the mutation showed a tendency towards an ERK signaling bias, which was not significant.

### 3.5. The Potent Synthetic Ligand NDP-α-MSH Changes the Signaling Bias of V103I and H158R

NDP-α-MSH is an improved synthetic form of α-MSH that has been shown to be more potent in G_S_ signaling [19]. Again, for *MC4R* V103I our results in G_S_ signaling were in line with most publications that used NDP-α-MSH [24,66]. For S127L, conflicting results for cAMP accumulation after NDP-α-MSH stimulation have been published with either an increase in EC_50_ [39,42] or a similar EC_50_ compared to the WT combined with a decreased E_max_ [41], or both an increase in EC_50_ and a diminished maximal response [40]. In our cell system, the mutation exhibited a similar signaling profile to the endogenous ligands. The response of H158R to NDP-α-MSH was not published yet, though again the observed gain-of-function similar to α-MSH was expected.

The synthetic ligand diminished the bias effect in the PLC of V103I and H158R, although for the MC4R WT the synthetic agonist was more potent in all pathways investigated. Interestingly, bias towards G_12/13_ and MAPK was increased for V103I. The latter has already been reported recently [23]. Here, it cannot result from G_q/11_ or G_i/o_ signaling, since these pathways are not increased. This hints towards an unknown mechanism independent of G proteins. For S127L, only one study investigated the ability to activate a non-G_S_ pathway. Here, the authors found that it could increase pERK levels when stimulated with NDP-α-MSH. Although it was only tested for one concentration of agonist and no conclusion about bias could be stated, the response was significantly lower than the MC4R WT [57]. In our hands, activation of ERK phosphorylation was also clearly impaired. The mutant H158R was investigated for ERK phosphorylation upon NDP-α-MSH stimulation. The response to a single concentration (1 µM) was similar to the MC4R WT [21], which is in contrast to our findings of an increased maximal response and a clear bias towards this pathway.

### 3.6. Summary of the Comprehensive Functional Characterization

This is the first in vitro study of *MC4R* mutations that determined the activation of all four G protein families in addition to ERK activation, β-arrestin2 recruitment, receptor expression and internalization. We could demonstrate that the signaling capacity of MC4R WT depends on its endogenous ligand. Our summarized data (Figure 7) show how each pathway was affected by the mutations. This study was the first to investigate the activation of G_12/13_ signaling after cell stimulation with endogenous ligands, although the overall activation is weak compared to other signaling pathways. So far, only one chimera-based study hinted towards such G protein coupling [11] and the physiological role at the MC4R still needs to be elucidated. Our study highlights that in order to understand how mutations might limit the function of MC4R in a physiological setting, both endogenous neuropeptides need to be investigated.

The results of the present study show how the *MC4R* mutations react differently to α- and β-MSH. Especially, the mutations V103I and H158R showed strong bias towards particular pathways, such as PLC activation, ERK phosphorylation and RhoA activation. The ligand NDP-α-MSH was highly potent only in G_S_ signaling, e.g., for V103I but not in activation of PLC where it had comparable potency to α- and β-MSH. These differences should be considered when *MC4R* mutations are characterized in vitro. In addition, the data gives an idea about the antagonistic potency of AgRP, though only one concentration of AgRP and one pathway has been investigated. For further studies, we would include all signaling pathways and perform a Schild’s analysis with increasing amounts of antagonist to gain further insight in the effect on the mutations. Evaluation of a complete set of signaling pathways offer new opportunities for the development of treatment options. This study demonstrates that a complete signaling profile of *MC4R* mutations, although necessarily done in an artificial cell system, may be helpful to estimate the pathophysiological role of *MC4R* mutations. For future studies, a cell system that is closer to the physiologic situation is urgently needed.

In conclusion, our data point out the complexity of MC4R signaling and its dependence on specific ligands. This implies that a cautious and standardized functional characterization is necessary to evaluate the functional impact of different *MC4R* mutations. This is of importance, as there is now growing evidence that more MC4R pathways are relevant for the development of obesity than previously expected.

## 4. Materials and Methods

### 4.1. Cell lines, Cloning and Reagents

Human embryonic kidney 293 (HEK293) cell line was purchased from ATCC, authenticated by single nucleotide polymorphism (SNP) analysis and regularly tested for mycoplasma contamination. Cells were cultivated in l-glutamine containing minimal essential medium (MEM, Biochrom, Berlin, Germany) supplemented with 5% fetal bovine serum (FBS, Gibco, Carlsbad, CA, USA) and 1% non-essential amino acids (NEA, Biochrom, Berlin, Germany) at 37 °C and humidified air containing 5% CO_2_. For cAMP measurements, reporter gene assays, total and cell surface expression (HiBiT assay, Promega, Mannheim, Germany), 1.5 × 10^4^ cells per well were seeded in poly-l-lysine-coated (Biochrom, Berlin, Germany) 96-well plates and incubated for 24h. The HiBiT assays were performed in white opaque, poly-l-lysine-coated 96-well plates (Corning, NY, USA #3917), and the other assays in translucent well plates (Falcon, Kaiserslautern, Germany). For Bioluminescence resonance energy transfer (BRET) assays, HEK293 cells were seeded in 6-well plates for transfection (8 × 10^5^ cells/well) and later reseeded in white opaque 96-well plates for measurement (2.2 × 10^4^ cells/well).

MC4R cDNA was amplified from genomic DNA, cloned into eukaryotic expression vector pcDps and *N*-terminally tagged with the hemagglutinin (5′ YPYDVPDYA 3′) epitope (HA). For β-arrestin recruitment BRET assays, the cDNA of rARRB2 (β-arrestin2, *Rattus norvegicus*) was cloned into the expression vector pFN31A (Promega, Mannheim, Germany), resulting in the *N*-terminal tag with NanoLuc (NL). MC4R and cholinergic muscarinic receptor 3 (rM3R, *R. norvegicus*) were *C*-terminally tagged with the protein tag Halotag (HT) using the pFC14A expression vector (Promega, Mannheim, Germany) when they serve as a mock control. For MC4R, the linker between receptor and protein tag was changed into ADPPVV using site-directed mutagenesis for improved assay performance. For total and cell surface expression, MC4R was cloned into pBiT3.1-N (Promega, Mannheim, Germany), resulting in an *N*-terminal tagged receptor with the HiBiT protein tag. The mutations V103I, S127L and H158R were incorporated into all expression vectors using site-directed mutagenesis. Mutagenesis primer can be found in Appendix A. All plasmids were sequenced and verified with BigDye-terminator sequencing (PerkinElmer Inc., Waltham, MA, USA) using an automatic sequencer (ABI 3710 XL; Applied Biosystems, Foster City, CA, USA). Protein tags do not affect MC4R functionality, which was tested via a cAMP accumulation assay (data not shown).

The compounds α-MSH, β-MSH, NDP-α-MSH, PTX and 3-Isobutyl-1-methylxanthine (IBMX) were purchased from Sigma-Aldrich (Darmstadt, Germany), and AgRP from Phoenix Pharmaceuticals (Karlsruhe, Germany).

### 4.2. Transfection

HEK293 cells were transfected 24 h after seeding. For cAMP measurements and reporter gene assays, cells were transfected with 45 ng plasmid DNA and 0.45 µL Metafectene (Biontex, Munich, Germany). In case of reporter gene assays, the transfection included an additional 45 ng of reporter DNA per well in MEM without supplements. For the HiBiT assay, transfection was performed as described previously [67].

In order to stain the receptor with fluorophore-tagged ligand, MC4R WT and the three mutations were transfected in HEK293 cells using Effectene^®^ (Quiagen, Hilden, Germany) transfection reagent according to the manufacturer’s instructions. HEK293 cells, seeded in 6-well plates (Sarstedt, Nürnbrecht, Germany) containing 24 mm #1 glass coverslips (Fisher Scientific, Hampton, NH, USA), were transfected with 0.4 µg DNA per well.

For BRET assays, HEK293 cells in 6 wells were transfected using 8 µL FuGene HD (Promega, Mannheim, Germany) and 2.4 µg DNA in Opti-MEM (Gibco) 4–6 h after seeding. The BRET partners were co-transfected with the NL:HT ratio of 1:10 (200 ng:2 µg). A total of 200 ng of pGEM-3Zf (+) was added as carrier DNA.

### 4.3. Determination of Total and Cell Surface Expression via HiBiT Assay

The number of receptors expressed on the cell membrane as well as total cell expression were determined using the NanoGlo^®^ HiBiT detection system (Promega, Mannheim, Germany). Here, HEK293 cells express the HiBiT-tagged receptor in low amounts and measurement was performed according to the manufacturer’s protocol (rapid measurements protocol). In short, 48 h after transfection, media was changed to 50 µL Opti-MEM without phenol red per well and injected with 50 µL of either HiBiT Extracellular substrate (Promega, Mannheim, Germany) for determination of cell surface expression or HiBiT Lytic substrate (Promega, Mannheim, Germany) for total expression using a plate reader (Mithras LB 940, Berthold Technologies GmbH & Co. KG, Bad Wildbad, Germany). Afterwards, the plate was shaken orbitally for 3 min at 300 cycles per minute, incubated for 10 min at room temperature and then luminescence was measured. Cells transfected with empty pcDNA3 were used as background control and were subtracted from the sample emissions.

### 4.4. Receptor Staining Using Fluorophor-Tagged Ligand

A total of 48 h after transfection and prior to imaging, cells were incubated with 100 nM TAMRA-NDP-*α*-MSH for 30 min and then washed with HBSS. Cells adhering to 24 mm #1 glass coverslips were mounted into Attofluor™ steel chambers (Thermofisher, Waltham, MA USA) and imaged in HBSS (Thermofisher, Waltham, MA, USA) using a SP8 (Leica, Wetzlar, Germany) inverted confocal microscope. The excitation wavelength was 561 nm (power at the sample of a few microwatts), and emitted fluorescence from TAMRA was collected in the 580–650 nm range using a Leica HyD Detector. The scan rate was 100 Hz, with a pixel size of the order of 50 nm.

Differential interference contrast (DIC) images were collected on a photo multiplier (PMT) placed on top of the condenser of the microscope.

### 4.5. Determination of β-Arrestin2 Recruitment via NanoBRET™

The recruitment of β-arrestin2 to MC4R was determined by a novel BRET technique, the NanoBRET™ (Promega, Mannheim, Germany), which utilizes an improved luciferase (NanoLuc, NL) as an energy donor and a protein tag (HaloTag, HT) that can bind to the energy acceptor (HT Ligand 618). The MC4R was *C*-terminally tagged with HT and β-arrestin2 was *N*-terminally tagged with NL. In case of a recruitment of β-arrestin2 to the receptor, which indicates the initiation of receptor internalization, radiation-free energy transfer occurs between energy donor and energy acceptor.

A total of 16–20 h after transfection in 6-well plates, cells were detached, and cell number was adjusted to 2.2 × 10^5^ cells/mL in Opti-MEM without phenol red supplemented with 4% FBS. Cells were divided into two pools, adding either the acceptor NanoBRET™ Ligand 618 (1 µL/mL cells of 0.1 mM solution) or the same amount of DMSO as background control. After reseeding 90 µL cell suspension/well into a white 96-well plates, incubation was continued for 4–6 h.

For BRET measurements, cells were challenged with 10 µL of either α-MSH, β-MSH or NDP-α-MSH (10 µM to 1 nM) for 5 min at 37 °C. For measurements, 25 µL of NanoBRET™ substrate was injected using a plate reader (Mithras LB 940). Donor and acceptor emission were measured at 460 nm and 610 nm, respectively, and the BRET ratio was calculated by dividing acceptor emission by donor emission:(1)$$\mathrm{BRET}\;\mathrm{ratio}\;=\frac{{\mathrm{emission}}_{\mathrm{acceptor}}}{{\mathrm{emission}}_{\mathrm{donor}}}$$

The ratio was corrected for background bleed through by subtracting of the background ratio (no acceptor DMSO control) and conversion to milliBRET units (mBU):(2)$$\mathrm{corrected}\;\mathrm{BRET}\;\mathrm{ratio}\;\left[\mathrm{mBU}\right]=({\mathrm{BRET}\;\mathrm{ratio}}_{\mathrm{sample}}-{\mathrm{BRET}\;\mathrm{ratio}}_{\mathrm{no}\;\mathrm{acceptor}\;\mathrm{control}})\times 1000$$

To calculate the difference between stimulated and basal interaction, the NET BRET was acquired using the formula:(3)$$\mathrm{NET}\;\mathrm{BRET}\;={\;\mathrm{corrected}\;\mathrm{BRET}\;\mathrm{ratio}}_{\mathrm{stim}}-{\;\mathrm{corrected}\;\mathrm{BRET}\;\mathrm{ratio}}_{\mathrm{basal}}$$

### 4.6. Determination of G_S_ and G_i/o_ Protein Coupling via Intracellular cAMP Accumulation

G_S_ and G_i_ coupling to the MC4R were determined using the AlphaScreen™ assay (Perkin Elmer Life Science, Boston, MA, USA) performed according to the manufacturer’s protocol and described elsewhere [68]. In brief, cells were stimulated 48h after transfection with either α-MSH, β-MSH, NDP-α-MSH (1 µM to 0.1 nM) or co-stimulation of these three ligands was performed with 100 µM AgRP in stimulation buffer containing IBMX for 40 min for G_S_ coupling. Determination of G_i_ coupling was investigated by pretreatment with 50 µg/mL PTX 18 h before stimulation. All stimulations were performed at 37 °C and 5% CO_2_. Incubation was stopped by removing the compounds and lysing the cells in lysis buffer containing IBMX for 1.5–2 h at 4 °C on a shaking platform.

The determination of cAMP using the AlphaScreen™ assay is a competitive immune assay based on antibody-coupled acceptor beads that bind cAMP and streptavidin-coupled donor beads that bind to a biotinylated cAMP probe. Excitement of the donor bead leads to the conversion of ambient oxygen to a singlet state. In close proximity of donor and acceptor beads through binding the biotinylated probe, the acceptor reacts with the singlet molecule and emits light. Absolute amounts of cAMP are determined by the simultaneous measurement of a cAMP standard. The plate reader (Mithras 940 LB)) was used for the read out, at 565 nm.

### 4.7. Determination of NFAT, RhoA Activation and ERK Phosphorylation via Reporter Gene Assays

Activation of PLC, RhoA and ERK1/2 phosphorylation was determined using luciferase-based reporter gene assays that utilize responsive elements in the promotor region of the gene encoding a firefly luciferase. For PLC activation, the NFAT responsive element (NFAT-luc, pGL4.33) was co-transfected with the receptor mutations. Similarly, co-transfections with SRF responsive element (SRF-luc, pGL4.33, Promega, Mannheim, Germany) for RhoA activation and Serum response element (SRE-luc, pGL3.4, Promega, Mannheim, Germany) for ERK1/2 phosphorylation were performed. A total of 48h post transfection, cells were challenged with α-MSH, β-MSH or NDP-α-MSH (10 mM to 1 nM) in MEM without supplements for 6 h at 37 °C and 5% CO_2_. The stimulation was stopped by discarding the media and cell lysis was induced by addition of 1× passive lysis buffer (PLB, Promega, Mannheim, Germany) and horizontal shaking for 15 min at room temperature.

Measurement of luciferase activity provided information about the activation of the respective second messenger by transferring 10 µL lysate into a white opaque 96-well plate. Injection of 40 µL firefly luciferase substrate (Promega, Mannheim, Germany) and measurement of luminescence was performed with a plate reader (Mithras LB 940).

### 4.8. Mathematical Models and Statistical Analysis

Statistical analysis was performed using two-way ANOVA, followed by Dunnett’s post hoc test in GraphPad Prism 6. Statistical significance was set at * *p* ≤ 0.05, ** *p* ≤ 0.01, *** *p* ≤ 0.001 and **** *p* ≤ 0.0001.

Concentration–response curves of each experiment were analyzed by fitting a non-linear regression model for sigmoidal response in Graph Pad Prism 6 (GraphPad Software Inc., La Jolla, CA, USA) to determine potency (EC_50_ values). For mutations that had EC_50_ values outside of concentration tested, it was set to the appropriate highest concentration tested. Signaling bias analysis was performed as described by Kenakin (2017) [69] using the formula
(4)$$\Delta \log(\frac{E_{max}}{EC_{50}})\;=\;\log\left(\frac{E_{\max test}}{EC_{50\;test}}\right)-\log\left(\frac{E_{\max ref}}{EC_{50\;ref}}\right)$$
with *test* referring to the tested receptor mutation and *ref* being reference (MC4R WT). E_max_ is the maximal response (efficiency) of the mutation, which was determined from the concentration–response curves. Where no E_max_ could be determined from incomplete curves (e.g., in case of AgRP co-stimulation), the response of the maximal concentration was substituted as has been described by Winpenny et al. (2016) [70].

Standard error (SEM) of Δlog(E_max_/EC_50_) was calculated using the equation:(5)$$SEM_{\Delta log}\;=\;\sqrt{{\left(SEM_{test}\right)}^{2}-{\left(SEM_{ref}\right)}^{2}}$$
where SEM*_test_* is the standard error the tested receptor mutation and SEM*_ref_* is the standard error of the reference (MC4R WT).

Bias was calculated creating the antilog of the Δlog (E_max_/EC_50_):(6)$$bias\;=10^{\Delta \log\left(\frac{E_{max}}{EC_{50}}\right)}$$

The standard error of the bias factor was determined by the equation:(7)$$SEM_{bias}\;=\;\frac{EC_{50}}{E_{max}}\times \frac{1}{EC_{50}}\times SEM_{E_{max}}+\frac{E_{max}}{EC_{50}}\times \frac{1}{E_{max}}\times SEM_{EC_{50}}$$

## Figures and Tables

**Figure 1 ijms-21-01224-f001:**
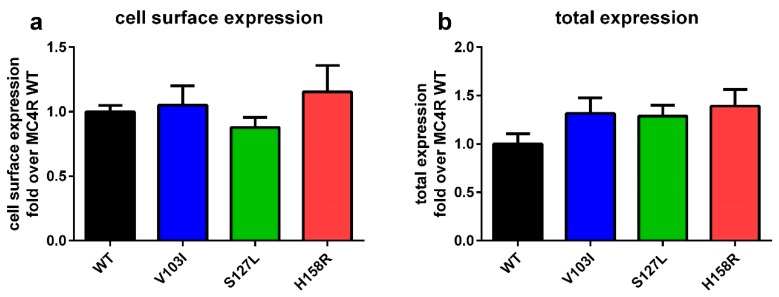
Cell surface expression and total expression of the melanocortin 4 receptor (*MC4R*) mutations do not significantly differ from each other. HEK293 cells were transfected with *N*-terminally HiBiT-tagged MC4R, a small protein tag, which is able to complement a split luciferase that cannot cross the plasma membrane. Either the cells membrane remained intact to determine receptors at the cell surface, or cells were lysed to determine the total expression of *MC4R* mutations. (**a**) Expression on the cell surface did not differ between the three *MC4R* mutations. (**b**) The total cell expression of the *MC4R* mutations was also not different. Values represent mean ± SEM from three to four independent experiments performed in triplicates. Statistical analysis was performed using a one-way ANOVA, followed by Dunnett’s post-hoc test.

**Figure 2 ijms-21-01224-f002:**
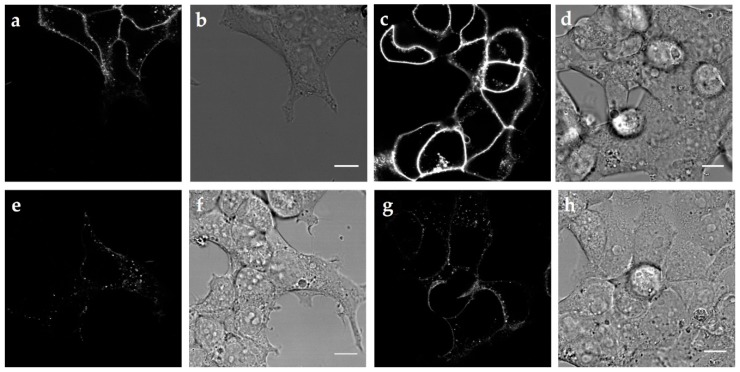
Confocal imaging of MC4R WT and three mutations labeled with TAMRA-NDP-*α*-MSH. (**a**) Confocal micrograph of cells expressing MC4R WT and (**b**) corresponding differential interference contrast (DIC) image. (**c**) Confocal micrograph of cells expressing MC4R V103I and (**d**) corresponding DIC image. (**e**) Confocal micrograph of cells expressing MC4R S127L and (**f**) corresponding DIC image. (**g**) Confocal micrograph of MC4R H158R and (**h**) corresponding DIC image. Scale bars are 10 µm. Images were acquired 30 min after labeling with 100 nM ligand, and subsequent washout. In panels (**a**,**c**,**e**,**g**) contrast was adjusted to the same overall intensity.

**Figure 3 ijms-21-01224-f003:**
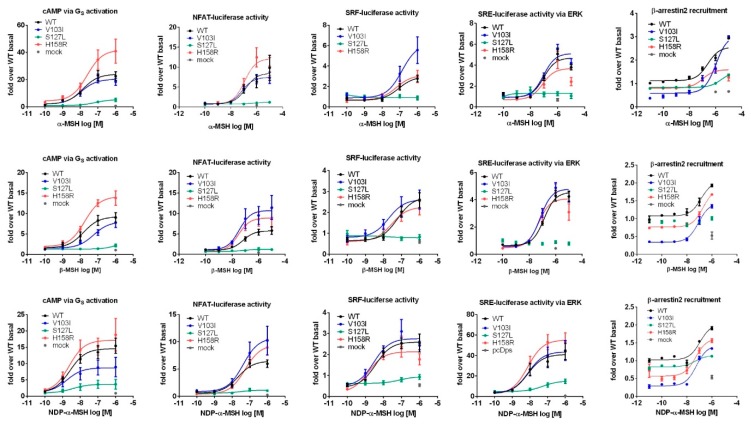
Concentration–response curves of all tested pathways were used to determine the potency (EC_50_) as well as the efficacy (E_max_) of each *MC4R* mutation in the respective pathway. All values were normalized to *MC4R* WT basal. The datasets contain pooled data from three to seven independent experiments each performed in triplicates. All concentration–response curves were analyzed with GraphPad Prism 6.0 using the non-linear regression model (sigmoidal response).

**Figure 4 ijms-21-01224-f004:**
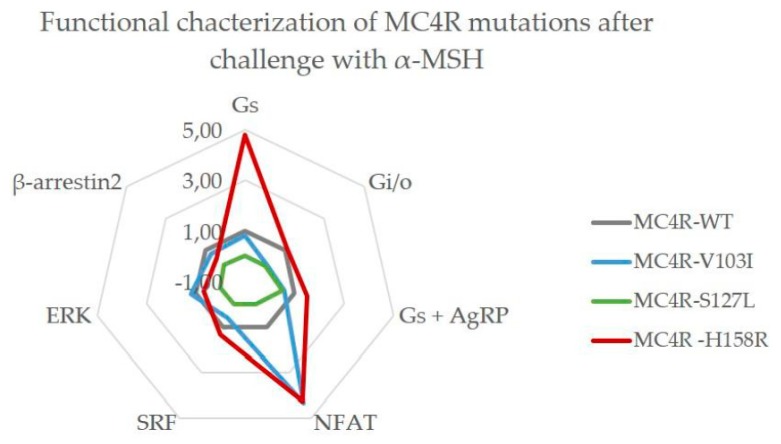
*MC4R* mutations V103I and H158R have a strong bias towards NFAT via PLC activation after challenge with α-MSH. As a reference, MC4R WT (gray) is plotted at a value of 1.0. The V103I mutant (blue) acted similarly to the WT in most pathways tested but had increased activity in the G_q/11_ pathway, and a decrease in G_i/o_ coupling. MC4R S127L (green) lost function in all tested pathways but responded to the antagonist Agouti related peptide (AgRP) in a similar manner to the WT. H158R (red) was a gain-of-function mutation for both the G_S_ and G_q/11_ pathway. Each value is based on EC_50_ and E_max_ values that were derived from non-linear regression models fitted to the concentration–response curves for each pathway from three to seven independent experiments performed in triplicates.

**Figure 5 ijms-21-01224-f005:**
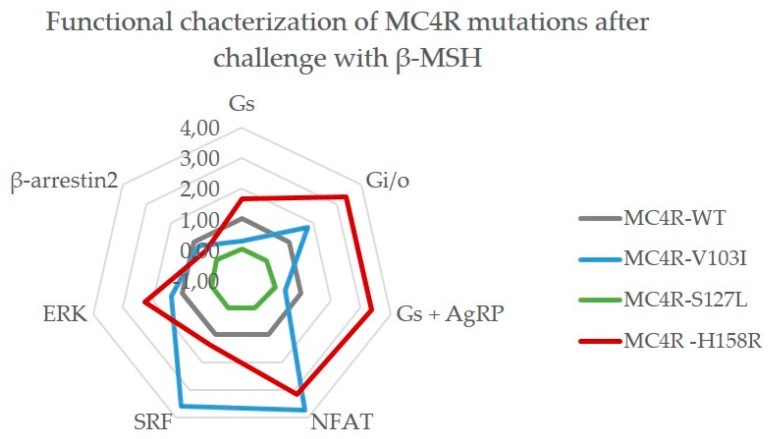
A β-MSH challenge created different responses for MC4R mutations V103I and H158R. As reference, MC4R WT (gray) is plotted at a value of 1.0. The V103I mutation (blue) showed impaired signaling activity in the G_S_ pathway with and without AgRP, but had a strong bias towards G_q/11_ (NFAT) and G_12/13_ (SRF), as well as towards G_i/o_ signaling. MC4R S127L (green) was impaired in all tested pathways. H158R (red) is a gain-of-function mutation for the G_S_, G_i/o_ and G_q/11_ pathway, as well as for ERK phosphorylation. Each value is based on EC_50_ and E_max_ values that were derived from non-linear regression models fitted to the concentration–response curves for each pathway from three to seven independent experiments performed in triplicates.

**Figure 6 ijms-21-01224-f006:**
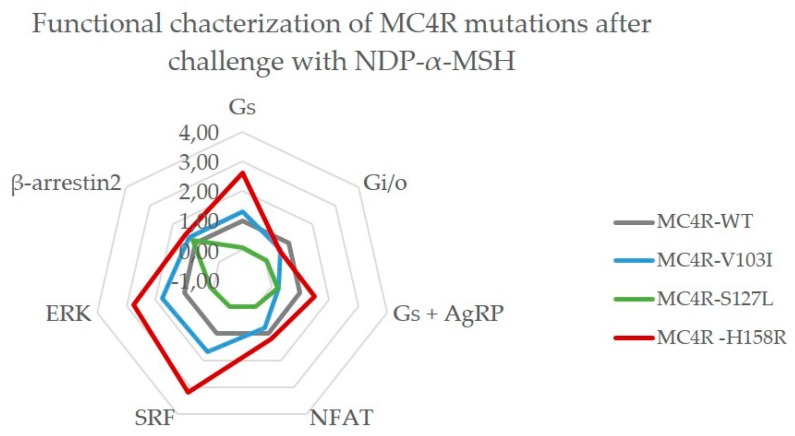
MC4R V103I and H158R lost their bias towards G_q/11_ after NDP-α-MSH stimulation. As a reference, MC4R WT (gray) is plotted at a value of 1.0. The MC4R V103I mutant (blue) lost the increased G_q/11_ signaling (NFAT) but gained a moderate bias towards G_12/13_ (SRF) and ERK phosphorylation. The loss-of-function mutation S127L (green) could not be rescued by NDP-α-MSH in most pathways, but β-arrestin2 recruitment was restored to WT-like. The H158R mutant (red) kept its gain-of-function in G_S_ signaling as was well as in G_12/13_ and ERK phosphorylation, at the same time it lost the strong bias towards G_q/11_ that was displayed with the α-MSH challenge. Each value is based on EC_50_ and E_max_ values that were derived from non-linear regression models fitted to the concentration–response curves for each pathway from three to seven independent experiments performed in triplicates.

**Figure 7 ijms-21-01224-f007:**
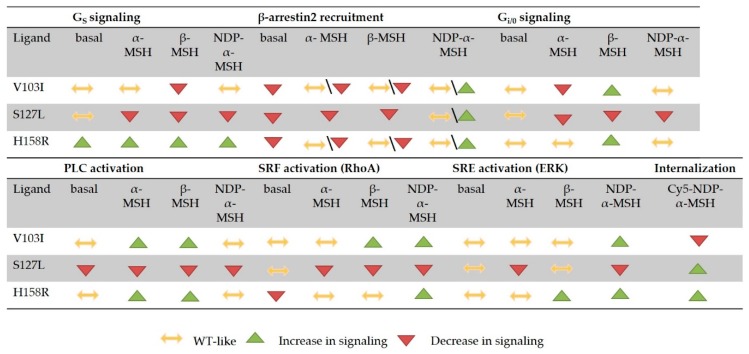
Summary of all functional data for the *MC4R* mutations V103I, S127L and H158R. This figure gives a crude overview of the differences in signaling profiles between the mutations, but also between the three tested ligands. Signaling is indicated as either decreased or increased when bias value deviated ≥0.5 bias values from the *MC4R* WT. For β-arrestin2 recruitment, the bias value differences were below 0.5, but statistically significant when analyzed with two-way ANOVA, followed by Dunnett’s post-hoc test.

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
