# Peer review of "Differential Signaling Profiles of MC4R Mutations with Three Different Ligands"

_ijms, 2020, doi:10.3390/ijms21041224_

Round 1

Reviewer 1 Report

In the manuscript entitled “Differential signaling profiles of MC4R mutations with three different ligands”, the authors reported a comprehensive study of the intracellular signalling activated by melanocortin 4 receptor (MC4R) in response to different ligands. They also evaluated the possible bias related to the introduction of three different mutations. The topic is interesting, the manuscript is well written and the results are well organized. However, there are some issues that the authors should consider before the manuscript is suitable for publication. In particular:

The main weakness of the manuscript is the validation of the used MC4R constructs. The authors reported several constructs used to evaluate different intracellular signalling. However, the effects of the different receptor modifications were not reported. The authors should demonstrate that the receptor modifications do not impact the physiological functionality almost of the WT receptor. The expression of MC4R and its mutant could be influenced by the introduction of a particular tag (HiBit). The authors should perform a western-blot analysis to demonstrate the expression of the MC4R used in the functional experiments by western-blot analysis. This aspect is critical to the interpretation of the obtained results. The authors should explain in the text why they evaluated only the effects on beta-arrestin2 recruitment as a marker of receptor internalization. The authors should introduce the statistical analysis of all the EC50 and Emax

Author Response

Please also see the attachment for figures.

In the manuscript entitled “Differential signaling profiles of MC4R mutations with three different ligands”, the authors reported a comprehensive study of the intracellular signalling activated by melanocortin 4 receptor (MC4R) in response to different ligands. They also evaluated the possible bias related to the introduction of three different mutations. The topic is interesting, the manuscript is well written and the results are well organized. However, there are some issues that the authors should consider before the manuscript is suitable for publication. In particular:
Point 1: The main weakness of the manuscript is the validation of the used MC4R constructs. The authors reported several constructs used to evaluate different intracellular signalling. However, the effects of the different receptor modifications were not reported. The authors should demonstrate that the receptor modifications do not impact the physiological functionality almost of the WT receptor. The expression of MC4R and its mutant could be influenced by the introduction of a particular tag (HiBit). The authors should perform a western-blot analysis to demonstrate the expression of the MC4R used in the functional experiments by western-blot analysis. This aspect is critical to the interpretation of the obtained results.

Response 1: We appreciate the comments of the reviewer. We agree that it is a critical point, whether the introduced modifications have an impact on receptor function. We are aware of the pitfalls of modifications within GPCRs with protein tags. We therefore performed cAMP accumulation assays with the tagged MC4R-WT for both tags the N-terminal HiBiT and the C-terminal HaloTag, to make sure that functionality of the receptor is not affected. We apologize for not having referenced these data in the initial manuscript. . There is no functional impairment of MC4R function due to the protein tags in terms of the main signaling pathway (Figure below). We added a reference to these data in line 448 – 449: “Protein tags do not affect MC4R functionality, which was tested via cAMP accumulation assay (data not shown).

Figure available in the attachment

Figure: Functional testing of protein-tagged MC4R. In order to investigate whether the N- and C-terminal protein tags affect the receptor signaling capacity, we investigated the ability of HiBiT-MC4R and MC4R-HaloTag to accumulate cAMP using the AlphaScreen Assay described in the manuscript. We performed a non-linear regression analysis using Graph Pad Prism 6 and compared the tagged versions with the untagged MC4R by performing an extra sum-of-squares F test, which resulted in one curve that fits all data sets.

Point 2: The authors should explain in the text why they evaluated only the effects on beta-arrestin2 recruitment as a marker of receptor internalization.
Response 2: The recruitment of β-arrestins after phosphorylation by GRKs (G protein coupled receptor kinases) initiates the internalization of GPCRs [1]. MC4R is part of the Class A of GPCRs, which is known for a higher affinity towards β-arrestin2 than β-arrestin1 [2]. Therefore, we particularly evaluated the recruitment of β-arrestin2.
For a long time, “β-arrestin-dependent”, “G protein-independent” signaling such as ERK phosphorylation was established for several GPCRs. In recent years, this dogma has been taken up by Grundmann et al., who used HEK293 cells which are lacking GS, Gq/11, G12/13 as well as a second group of cells lacking both β-arrestins. Both cell types were treated with PTX to block Gi/o, and/or a specific blocker for Gq/11 signaling (FR900359) to compare ERK signaling in “zero functional G” vs. “zero arrestin” cells. Surprisingly, several GPCRs in cells with “zero functional G” were not able to induce ERK signaling, whereas arrestin-devoid cells were not severely affected in this ability. The authors therefore concluded that the function of arrestins in ERK phosphorylation might be more related to scaffolding and not to the initiation of signaling [3]. Lutrell et al. argued that the deletion of all functional G proteins in the cell might change their signaling behavior due to a selective pressure after comparing the ERK phosphorylation in CRISPR/Cas9-edited cells to the parental HEK293 cell line [4]. Currently it is not yet clear and there is a controversial debate as to whether there is "β-arrestin-dependent" signaling or not. We defined β-arrestin2 recruitment as part of the internalization machinery as stated in lines 265 – 269. Additionally, we observed internalization microscopically using fluorophore-coupled NDP-α-MSH and counted formed endosomes in Figure 2 and Table S1.
Point 3: The authors should introduce the statistical analysis of all the EC50 and Emax.

Response 3: The EC50 and Emax values have now been statistically analysed by using a One-way ANOVA for Emax value comparison and a Two-way ANOVA for EC50 value comparison, both followed by Dunnett’s post-hoc test. This part has been modified in the manuscript.

1.Benovic, J. L.; Kühn, H.; Weyand, I.; Codina, J.; Caron, M. G.; Lefkowitz, R. J., Functional desensitization of the isolated beta-adrenergic receptor by the beta-adrenergic receptor kinase: potential role of an analog of the retinal protein arrestin (48-kDa protein). Proceedings of the National Academy of Sciences 1987, 84, (24), 8879-8882.
2.Oakley, R. H.; Laporte, S. A.; Holt, J. A.; Caron, M. G.; Barak, L. S., Differential affinities of visual arrestin, βarrestin1, and βarrestin2 for G protein-coupled receptors delineate two major classes of receptors. Journal of Biological Chemistry 2000, 275, (22), 17201-17210.
3.Grundmann, M.; Merten, N.; Malfacini, D.; Inoue, A.; Preis, P.; Simon, K.; Rüttiger, N.; Ziegler, N.; Benkel, T.; Schmitt, N. K.; Ishida, S.; Müller, I.; Reher, R.; Kawakami, K.; Inoue, A.; Rick, U.; Kühl, T.; Imhof, D.; Aoki, J.; König, G. M.; Hoffmann, C.; Gomeza, J.; Wess, J.; Kostenis, E., Lack of beta-arrestin signaling in the absence of active G proteins. Nature Communications 2018, 9, (1), 341.
4.Luttrell, L. M.; Wang, J.; Plouffe, B.; Smith, J. S.; Yamani, L.; Kaur, S.; Jean-Charles, P.-Y.; Gauthier, C.; Lee, M.-H.; Pani, B.; Kim, J.; Ahn, S.; Rajagopal, S.; Reiter, E.; Bouvier, M.; Shenoy, S. K.; Laporte, S. A.; Rockman, H. A.; Lefkowitz, R. J., Manifold roles of β-arrestins in GPCR signaling elucidated with siRNA and CRISPR/Cas9. Science Signaling 2018, 11, (549), eaat7650.

Reviewer 2 Report

In this manuscript by Paisdzior et al., the authors investigate melanocortin 4 receptor polymorphic mutations found in the human population and comprehensively characterize these mutants at several different signaling pathways.

Although I find the analysis interesting and with great interest in the field of obesity and anorexia behaviors, I find several issues and discrepancies that should be addressed, likely with additional experiments.

Major Points:

A recent Cell paper (Lotta et al. 2019) also shows comparisons of the b-arrestin recruitment pathway between WT and V103I mutant, where the Cell paper shows an increase in b-arrestin recruitment, contrary to what this manuscript reports. Both groups are using a similar NanoBRET assay for measuring b-arrestin recruitment, however the authors in this manuscript state, “This discrepancy might be explained by the way of data generation, as we did concentration response curves.”

However, this is misleading because the authors in the Lotta et al. paper also did concentration response curves (see Fig 4B) and appear to normalize the data similarly with respect to WT and yet very clearly see an increase in b-arrestin recruitment.

How do the authors reconcile these differences (which appear to be very important with respect to this mutation)? Of note, I also see in this current manuscript by Paisdzior et al.,  b-arrestin recruitment for WT appears to be an order of magnitude weaker than what is reported in the Lotta et al. paper. The authors should try to replicate the data in the Lotta et al. paper with similar conditions. I also recommend to explore the phosphorylation state of the receptor with respect to which G protein Receptor Kinase (GRK) may be involved, because there could be differences here in the rate-limiting step of b-arrestin recruitment. Also to look at several time points of incubation as b-arrestin recruitment can be dynamic over time.

Authors state Gi coupling was determined using pertussis toxin (PTX) to uncouple Gi/o proteins, but it appears the authors are still measuring cAMP accumulation and not inhibition, which is what Gi/o proteins canonically function to do via inhibition of adenylyl cyclase (AC). Can the authors clarify what they are measuring with respect to cAMP and Gi/o activity?

If I am understanding their methods correctly, they are measuring accumulation with and without PTX, which to me is an indirect way to measure Gi/o coupling because in either instance the cAMP accumulation is still measuring Gs activity--- they can only measure influence of Gi/o proteins on cAMP accumulation via with PTX treated cells, but yet it is still Gs activity. Instead, they should be measuring Gi/o-mediated cAMP inhibition directly using usually forskolin stimulated AC with and without PTX, which would directly be measuring Gi/o coupling activity.

The authors state on several occasions that the AGRP antagonist activity appeared to be affected. Unfortunately, this is impossible to quantify by only testing one concentration of AGRP in the presence of agonist. Two options to measure if AGRP antagonist activity is affected are to 1) Measure changes in affinity using radioligand binding or another binding method OR 2) Use a Schild analysis varying the concentration of AGRP on agonist stimulation to calculate a Kb for AGRP and compare amongst mutants. Option 1 is of course can be expensive and literally will not tell much about changes to the antagonist action on any given activation pathway, therefore Option 2 is likely to yield key results for AGRP antagonist activity changes for every respective pathway. See Kenakin and Beek 1981 JPET for Schild analysis. Although the analysis using luciferase reporters (e.g. SRE, NFAT, SRF) is important, I think these may also reflect other downstream effectors rather than titled/labeled as “PLC activation” or “RhoA activation.” Please label the graphs and radar plots as exactly what is measured (SRE-luciferase). Unless the authors do controls for each of these assays (for instance, knockdown/inhibition of Gq/11 proteins for loss of NFAT activity, similarly for G12/13 on SRF), then it is unknown if there are several effectors influencing these responses. For example and something for the authors to note, that SRE luciferase activity via ERK could be due to several effectors (Gs, Gq/11, or G12/13 even b-arrestin). Therefore, without the knockdown/knockout/inhibition controls, it is not clear which effectors are involved in these luciferase reporter responses. A key recommendation for determination of bias, independent of the analysis as shown in the radar plots, is to show the concentration response overlay data found in Fig S1 in Fig 3, mainly because there is a lot of information in the concentration response curves, which in fact as the authors have pointed out with respect to the Lotta et al. Cell paper, this is key information for all mutants.

Author Response

In this manuscript by Paisdzior et al., the authors investigate melanocortin 4 receptor polymorphic mutations found in the human population and comprehensively characterize these mutants at several different signaling pathways.

Although I find the analysis interesting and with great interest in the field of obesity and anorexia behaviors, I find several issues and discrepancies that should be addressed, likely with additional experiments.

Major Points:

Point 1: A recent Cell paper (Lotta et al. 2019) also shows comparisons of the b-arrestin recruitment pathway between WT and V103I mutant, where the Cell paper shows an increase in b-arrestin recruitment, contrary to what this manuscript reports. Both groups are using a similar NanoBRET assay for measuring b-arrestin recruitment, however the authors in this manuscript state, “This discrepancy might be explained by the way of data generation, as we did concentration response curves.” However, this is misleading because the authors in the Lotta et al. paper also did concentration response curves (see Fig 4B) and appear to normalize the data similarly with respect to WT and yet very clearly see an increase in b-arrestin recruitment. How do the authors reconcile these differences (which appear to be very important with respect to this mutation)? Of note, I also see in this current manuscript by Paisdzior et al.,  b-arrestin recruitment for WT appears to be an order of magnitude weaker than what is reported in the Lotta et al. paper. The authors should try to replicate the data in the Lotta et al. paper with similar conditions.

Response 1: We thank the reviewer for the comprehensive review of our manuscript. We appreciate this comment. We would like to point out that we used a different method than the research group in the Cell paper (Lotta et al. 2019). In the paper from Lotta and coworkers, a NanoBiT assay was utilized. Although both NanoBiT and NanoBRET use the NanoLuc luciferase, the NanoBiT assay is based on a split-luciferase method, whereas the NanoBRET method we used, is a protein-protein interaction method based on the energy transfer from a donor luciferase to an acceptor a fluorophore when the interaction partners are in close proximity. For the measurement of β-arrestin recruitment both, the BRET and the split-luciferase approaches have been published. Since the BRET method is the most established method to demonstrate β-arrestin recruitment in many studies [1-3], we took advantage of this procedure. For this reason, a direct comparison of the results in the Lotta et al. paper and our study is difficult.

The publication by Lotta et al. contained concentration-response curves, but these were normalized to the maximum response of the WT, compared to a fold over WT basal normalization, as we presented in the manuscript. Due to the different measurement approaches and normalization processes, the results cannot be easily compared and a judgement of weaker or stronger recruitment is not possible. Based on the comments, we modified the lines 296 - 297 to “…, as we utilized a bioluminescence resonance energy transfer method (NanoBRET) in comparison to the split-luciferase approach.”

The most apparent difference between the β-arrestin data of the Cell paper and our data is the constitutive recruitment of the V103I mutant. In our hands, the recruitment in a basal state was reduced to 38% of the WT signal, whereas Lotta et al. had an increased basal recruitment of approximately 80% of WT maximum activation.

Also, the disturbance of internalization of the mutation has been reported in our and the Lotta et al. study - again using a similar, but not identical method (Fig 2). Considering β-arrestins to be an important part of the internalization machinery, a decrease of basal recruitment would explain the impairment in V103I internalization (less internalization as shown in the Lotta et al. paper), whereas an increase as reported in the Lotta et al., paper would be expected to increase internalization. Similarly, we determined the tendency of V103I to increased ERK phosphorylation as it was reported. In addition, Lotta et al. performed concentration-response-experiments to measure cAMP accumulation, where they showed a clear gain-of-function for V103I. This is in conflict with the currently available literature [4-9], in which V103I was reported to behave like WT. It is already reported that conflicting in vitro data concerning characterization of GPCR mutations are published, as for example in case of the mutation S127L ([10-12] vs. [13]). However, with the exception of the recent findings of β-arrestin2 recruitment, our data are in accordance with current data in the literature for V103I. For now, we cannot reconcile the differences in β-arrestin2 recruitment between the Lotta et al. and our data, though we are aware that the evaluation might be of prime importance for future studies.

Point 2: I also recommend to explore the phosphorylation state of the receptor with respect to which G protein Receptor Kinase (GRK) may be involved, because there could be differences here in the rate-limiting step of b-arrestin recruitment. Also to look at several time points of incubation as b-arrestin recruitment can be dynamic over time.

Response 2: We agree with the reviewer that this is an excellent idea. The aim of our study was a comprehensive characterization of three selected MC4R mutations in terms of G protein activation in addition to the current expression and internalization studies. The investigation of phosphorylation to analyze which GRK might be involved is part of future studies. We added to the discussion part of the manuscript: “To investigate the conflicting evidence for V103I, future approaches could involve a kinetic measurement of β-arrestin2 recruitment as well as phosphorylation by GPCR kinases (GRKs).” (lines 298 – 299).

Point 3: Authors state Gi coupling was determined using pertussis toxin (PTX) to uncouple Gi/o proteins, but it appears the authors are still measuring cAMP accumulation and not inhibition, which is what Gi/o proteins canonically function to do via inhibition of adenylyl cyclase (AC). Can the authors clarify what they are measuring with respect to cAMP and Gi/o activity? If I am understanding their methods correctly, they are measuring accumulation with and without PTX, which to me is an indirect way to measure Gi/o coupling because in either instance the cAMP accumulation is still measuring Gs activity--- they can only measure influence of Gi/o proteins on cAMP accumulation via with PTX treated cells, but yet it is still Gs activity. Instead, they should be measuring Gi/o-mediated cAMP inhibition directly using usually forskolin stimulated AC with and without PTX, which would directly be measuring Gi/o coupling activity.

Response 3: We are sorry that our approach of determination of Gi activity was not clearly explained. It is correct that using PTX leads to an indirect measurement of Gi/o activation. As PTX locks Gαi/o in its GDP-bound and therefore inactivated state, the G protein is no longer able to inhibit adenylyl cyclase (AC) to form cAMP [14]. Gi/o activation is determined as activation of AC in the presence of PTX subtracted from AC activation in the absence of PTX. In both cases, cAMP accumulation is the read-out system. So far, we used the raw data including the least possible amount of deduction and calculation. Therefore, we chose plots that show the increase of cAMP production after PTX treatment. The difference between PTX-treated and non-treated cells can be linked to the cAMP production that was inhibited by Gi/o activity, which results in an indirect way of measurement.

It has come to our attention that the treatment with forskolin increases the affinity of AC towards GαS [15, 16], which makes the investigation of the inhibition of cAMP production difficult, when the receptor is coupled to both GS and Gi/o. We therefore decided to use the aforementioned method.

Point 4: The authors state on several occasions that the AGRP antagonist activity appeared to be affected. Unfortunately, this is impossible to quantify by only testing one concentration of AGRP in the presence of agonist. Two options to measure if AGRP antagonist activity is affected are to 1) Measure changes in affinity using radioligand binding or another binding method OR 2) Use a Schild analysis varying the concentration of AGRP on agonist stimulation to calculate a Kb for AGRP and compare amongst mutants. Option 1 is of course can be expensive and literally will not tell much about changes to the antagonist action on any given activation pathway, therefore Option 2 is likely to yield key results for AGRP antagonist activity changes for every respective pathway. See Kenakin and Beek 1981 JPET for Schild analysis.

Response 4: We see that the reviewer has a good point and appreciate the ideas that were presented. Unfortunately, a Schild analysis of all activation pathways with different concentration of AgRP is out of the scope of this particular research manuscript. However, we added this as a future perspective to the discussion: “In addition, the data gives an idea about the antagonistic potency of AgRP, although only one concentration of AgRP and one pathway has been investigated. For further studies, we would include all signaling pathways and perform a Schild’s analysis with increasing amounts of antagonist to gain further insight in the effect on the mutations.” (lines 403 – 406).

Point 5: Although the analysis using luciferase reporters (e.g. SRE, NFAT, SRF) is important, I think these may also reflect other downstream effectors rather than titled/labeled as “PLC activation” or “RhoA activation.” Please label the graphs and radar plots as exactly what is measured (SRE-luciferase). Unless the authors do controls for each of these assays (for instance, knockdown/inhibition of Gq/11 proteins for loss of NFAT activity, similarly for G12/13 on SRF), then it is unknown if there are several effectors influencing these responses. For example and something for the authors to note, that SRE luciferase activity via ERK could be due to several effectors (Gs, Gq/11, or G12/13 even b-arrestin). Therefore, without the knockdown/knockout/inhibition controls, it is not clear which effectors are involved in these luciferase reporter responses.

Response 5: We agree with the reviewer and changed labeling within the manuscript, in particular Figures 3,4,5,6 and S1. We are aware of the fact that ERK phosphorylation can be the result of several signaling cascades as the reviewer stated. The luciferase-activity measured via the SRE reporter is nevertheless a direct consequence of phosphorylated ERK activating luciferase expression. In addition, its close name resemblance to the SRF-reporter, we kept the ERK as synonym for SRE in the radar plots and edited it to “SRE-luciferase activity via ERK” in the concentration-response curves in Figure S1.

Point 6: A key recommendation for determination of bias, independent of the analysis as shown in the radar plots, is to show the concentration response overlay data found in Fig S1 in Fig 3, mainly because there is a lot of information in the concentration response curves, which in fact as the authors have pointed out with respect to the Lotta et al. Cell paper, this is key information for all mutants.

Response 6: We agree with the reviewer that the concentration-response curves are in fact the key information of our work. Although MC4R mutation bias has been reported before, this is the first article collecting data for three different mutations to this extend. We felt that the amount of graphs in Fig S1 are overwhelming and information can easily get lost, therefore we opted for the more comprehendible radar plots. To add transparency and further information to the reader, we listed all measured values (basal, EC50, Emax, bias value) as well as included the concentration-response curves in the supplement.

1.Breit, A.; Wolff, K.; Kalwa, H.; Jarry, H.; Büch, T.; Gudermann, T., The natural inverse agonist agouti-related protein induces arrestin-mediated endocytosis of melanocortin-3 and-4 receptors. Journal of Biological Chemistry 2006.
2.Kocan, M.; See, H. B.; Seeber, R. M.; Eidne, K. A.; Pfleger, K. D. G., Demonstration of Improvements to the Bioluminescence Resonance Energy Transfer (BRET) Technology for the Monitoring of G Protein–Coupled Receptors in Live Cells. Journal of Biomolecular Screening 2008, 13, (9), 888-898.
3.Smith, J. S.; Alagesan, P.; Desai, N. K.; Pack, T. F.; Wu, J.-H.; Inoue, A.; Freedman, N. J.; Rajagopal, S., C-X-C Motif Chemokine Receptor 3 Splice Variants Differentially Activate Beta-Arrestins to Regulate Downstream Signaling Pathways. Molecular Pharmacology 2017, 92, (2), 136-150.
4.Gu, W.; Tu, Z.; Kleyn, P. W.; Kissebah, A.; Duprat, L.; Lee, J.; Chin, W.; Maruti, S.; Deng, N.; Fisher, S. L., Identification and functional analysis of novel human melanocortin-4 receptor variants. Diabetes 1999, 48, (3), 635-639.
5.Ho, G.; MacKenzie, R. G., Functional characterization of mutations in melanocortin-4 receptor associated with human obesity. Journal of Biological Chemistry 1999, 274, (50), 35816-35822.
6.Rovite, V.; Petrovska, R.; Vaivade, I.; Kalnina, I.; Fridmanis, D.; Zaharenko, L.; Peculis, R.; Pirags, V.; Schioth, H. B.; Klovins, J., The role of common and rare MC4R variants and FTO polymorphisms in extreme form of obesity. Molecular Biology Reports 2014, 41, (3), 1491-1500.
7.Hughes, D. A.; Hinney, A.; Brumm, H.; Wermter, A.-K.; Biebermann, H.; Hebebrand, J.; Stoneking, M., Increased constraints on MC4R during primate and human evolution. Human Genetics 2008, 124, (6), 633.
8.Melchior, C.; Schulz, A.; Windholz, J.; Kiess, W.; Schöneberg, T.; Körner, A., Clinical and Functional Relevance of Melanocortin-4 Receptor Variants in Obese German Children. Hormone Research in Paediatrics 2012, 78, (4), 237-246.
9.Thearle, M. S.; Muller, Y. L.; Hanson, R. L.; Mullins, M.; AbdusSamad, M.; Tran, J.; Knowler, W. C.; Bogardus, C.; Krakoff, J.; Baier, L. J., Greater Impact of Melanocortin-4 Receptor Deficiency on Rates of Growth and Risk of Type 2 Diabetes During Childhood Compared With Adulthood in Pima Indians. Diabetes 2012, 61, (1), 250-257.
10.Lubrano-Berthelier, C.; Durand, E.; Dubern, B.; Shapiro, A.; Dazin, P.; Weill, J.; Ferron, C.; Froguel, P.; Vaisse, C., Intracellular retention is a common characteristic of childhood obesity-associated MC4R mutations. Human Molecular Genetics 2003, 12, (2), 145-153.
11.Valli-Jaakola, K.; Lipsanen-Nyman, M.; Oksanen, L.; Hollenberg, A. N.; Kontula, K.; Bjørbæk, C.; Schalin-Jäntti, C., Identification and Characterization of Melanocortin-4 Receptor Gene Mutations in Morbidly Obese Finnish Children and Adults. The Journal of Clinical Endocrinology & Metabolism 2004, 89, (2), 940-945.
12.Xiang, Z.; Litherland, S. A.; Sorensen, N. B.; Proneth, B.; Wood, M. S.; Shaw, A. M.; Millard, W. J.; Haskell-Luevano, C., Pharmacological Characterization of 40 Human Melanocortin-4 Receptor Polymorphisms with the Endogenous Proopiomelanocortin-Derived Agonists and the Agouti-Related Protein (AGRP) Antagonist. Biochemistry 2006, 45, (23), 7277-7288.
13.Hinney, A.; Hohmann, S.; Geller, F.; Vogel, C.; Hess, C.; Wermter, A.-K.; Brokamp, B.; Goldschmidt, H.; Siegfried, W.; Remschmidt, H., Melanocortin-4 receptor gene: case-control study and transmission disequilibrium test confirm that functionally relevant mutations are compatible with a major gene effect for extreme obesity. The Journal of Clinical Endocrinology & Metabolism 2003, 88, (9), 4258-4267.
14.Mangmool, S.; Kurose, H., Gi/o protein-dependent and-independent actions of pertussis toxin (PTX). Toxins 2011, 3, (7), 884-899.
15.Dessauer, C. W.; Scully, T. T.; Gilman, A. G., Interactions of forskolin and ATP with the cytosolic domains of mammalian adenylyl cyclase. Journal of Biological Chemistry 1997, 272, (35), 22272-22277.
16.Bräunig, J.; Mergler, S.; Jyrch, S.; Hoefig, C. S.; Rosowski, M.; Mittag, J.; Biebermann, H.; Khajavi, N., 3-Iodothyronamine Activates a Set of Membrane Proteins in Murine Hypothalamic Cell Lines. Frontiers in Endocrinology 2018, 9, (523).

Round 2

Reviewer 2 Report

The author's have addressed many of my concerns. I have the following recommendations for publication:

Related to point 3- Please change in the figures and text from "Gi/o coupling" to "contributions from Gi/o coupling" considering this is an indirect measure of Gi/o activity and not a direct measurement of activity.

Related to point 6- I strongly recommend including key dose-response curves in the main figures. Again this conveys the major findings, especially in light of the Lotta et al. paper, that should be displayed.

Author Response

The author's have addressed many of my concerns. I have the following recommendations for publication:

Related to point 3- Please change in the figures and text from "Gi/o coupling" to "contributions from Gi/o coupling" considering this is an indirect measure of Gi/o activity and not a direct measurement of activity.

Response: We appreciate the comment of the reviewer. It is of course correct that determination of Gi/o activity as a result adenylyl cyclase (AC) activity in the absence or presence of PTX is indirect. Nevertheless, we directly determine the second messenger (cAMP) that is modulated by Gi/o activity at AC. The amount of cAMP determined without PTX is the result of both the activation of AC by GS and the inhibition by Gi/o. We would prefer to address this issue in the introductory part of the results in lines 94 – 99: “…we measured Gi/o activation via determination of second messenger (cAMP) modulation by AC in the absence or presence of pertussis toxin (PTX) pretreatment. In the presence of PTX, which locks the Gαi/o subunit in its GDP-bound inactivated state, the G protein is no longer able to inhibit the AC to form cAMP [48]. Although this is an indirect way of measuring the contributions Gi/o signaling to the total amount of intracellular cAMP, we will refer to these measurements Gi/o coupling in order to simplify the results part.” We think this solution will keep the readability of the manuscript on its current level.

Related to point 6- I strongly recommend including key dose-response curves in the main figures. Again this conveys the major findings, especially in light of the Lotta et al. paper, that should be displayed.

Response: We now included the concentration-response-curves as figure 3 in the main text.